# Recent Progress of Low and Medium-Carbon Advanced Martensitic Steels

Koh-ichi Sugimoto 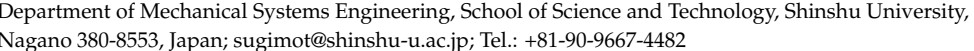

Department of Mechanical Systems Engineering, School of Science and Technology, Shinshu University, Nagano 380-8553, Japan; sugimot@shinshu-u.ac.jp; Tel.: +81-90-9667-4482

**Abstract:** This article introduces the microstructural and mechanical properties of low and medium-carbon advanced martensitic steels (AMSs) subjected to heat-treatment, hot- and warm- working, and/or case-hardening processes. The AMSs developed for sheet and wire rod products have a tensile strength higher than 1.5 GPa, good cold-formability, superior toughness and fatigue strength, and delayed fracture strength due to a mixture of martensite and retained austenite, compared with the conventional martensitic steels. In addition, the hot- and warm-stamping and forging contribute to enhance the mechanical properties of the AMSs due to grain refining and the improvement of retained austenite characteristics. The case-hardening process (fine particle peening and vacuum carburization) is effective to further increase the fatigue strength.

**Keywords:** advanced martensitic steel; retained austenite characteristics; microstructure; mechanical properties; heat treatment; hot-stamping; hot-forging; case hardening

## 1. Introduction

The strain-induced transformation of austenite to martensite enhances the ductility of austenitic steels such as Fe-Ni, Fe-Ni-C, and Fe-Cr-Ni steels. These high-alloy austenitic steels are called TRansformation-Induced Plasticity (TRIP) steels [1,2]. In the 1980s, low and medium carbon Si-Mn ferritic steels subjected to intercritical annealing and then isothermal transformation (IT) or austempering process were developed by Sakuma et al. [3,4]. The steel is named low alloy TRIP-aided steel or TRIP-assisted steel because it achieves high ductility by the TRIP effect of metastable retained austenite of 5 to 30 vol %. The TRIP-aided steel was mainly applied to the automotive body parts that need high cold press formability and weldability [4–6]. Up to now, various kinds of low and medium carbon advanced ultrahigh- and high-strength steels (AHSSs) with metastable retained austenite of different volume fraction, stability, size, morphology, and chemical composition were developed for the weight reduction and the improvement of crash safety of the automotive body [7–10].

In general, the AHSSs are categorized as the following: first-, second- and third-generation AHSSs [7–9]. The second-generation AHSSs are high Mn austenitic steels with an Mn content higher than 14 mass % and are named TWinning-Induced Plasticity (TWIP) steels [11]. The first- and third-generation AHSSs except for medium Mn (MMn) steels with 4 to 12% Mn are lean micro-alloyed Si/Al-Mn steels. The third-generation AHSSs are classified into two types, Type A and Type B, by the kind of matrix structure.

(I). First-generation AHSS: ferrite–martensite dual-phase (DP) steel [7,9,12–16], TRIP-aided polygonal ferrite (TPF) steel [3–7,9,17–19], TRIP-aided annealed martensite (TAM) steel [20–22], and complex-phase (CP) steel [7,9,15,23],

(II). Second-generation AHSS: high Mn TWIP and TWIP/TRIP steels [7,9,11,24–28],

(III). Third-generation AHSS (Type A): TRIP-aided bainitic ferrite (TBF) steel [9,10,29–33], one-step and two-step quenched and partitioned (Q&P) steels [7–9,25,34–41], carbide-free bainitic (CFB) steel [42–49], and duplex type medium manganese (D-MMn) steel [9,25,50–57].

(IV). Third-generation AHSS (Type B): TRIP-aided martensitic (TM) steel [58–63] and martensite-type medium manganese (M-MMn) steel [53,64–69], which are called advanced martensitic steel (AMS).

The product of tensile strength and total elongation (TS×TEl) of various AHSSs as a function of austenite or retained austenite fraction is shown in Figure 1. For the third-generation AHSSs (Type A), a TS×TEl higher than 30 GPa% is required to apply to the automotive body frame members, the seat frame members, the concrete mixer truck cylinders, etc. In connection with this, the IT process at low temperatures during martensite-start temperature ($M_s$) and martensite-finish temperature ($M_f$) is recently applied to TBF, one-step Q&P, and CFB steels [30–32,34–41,43,47,48,58–61], which achieve high tensile strength and mechanical properties due to bainitic ferrite/martensite (BF/M) structure matrix. To obtain the tensile strength higher than 1.5 GPa, TM and M-MMn steels with martensitic structure matrix are recently developed [58–69]. These steels are classified as the third-generation AHSSs (Type B). Hereafter, these third-generation AHSSs (type B) are also called low and medium-carbon "*Advanced Martensitic Steel* (AMS)", because the martensitic structure is the main matrix structure.

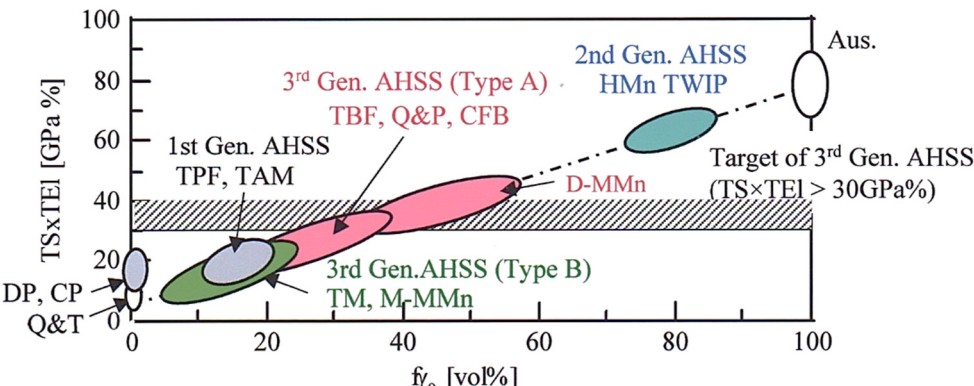

**Figure 1.** Relationship between the product of tensile strength and total elongation (TS×TEl) and initial volume fraction of austenite or retained austenite ($f\gamma_0$) in the first-, second-, and third-generation (Type A and Type B) advanced high-strength steels (AHSSs). Q&T: conventional quenched and tempered martensitic steel, DP: ferrite–martensite dual-phase steel, CP: complex-phase steel, TPF, TAM, TBF, and TM: transformation-induced plasticity (TRIP)-aided steels with polygonal ferrite, annealed martensite, bainitic ferrite, and martensite matrix structure, respectively. Q&P: one-step and two-step quenched and partitioned steel, CFB: carbide-free bainitic steel, D-MMn: duplex-type medium Mn steel, M-MMn: martensite-type medium Mn steel, HMn TWIP: high manganese TWIP steel, Aus: austenitic steel. This figure is reproduced based on Ref. [52]. Reprinted with permission from Elsevier: Mater. Sci. Eng. A, Copyright 2021.

To produce the AMS, two kinds of heat-treatment process are proposed: (1) IT process below $M_f$ [59–63] and (2) direct quenching to room temperature (DQ) [58–63,70–73]. In the case of $M_f$ > room temperature, the partitioning process will be added after the DQ process. The AMS exhibits higher TS×TEl [59–63] (Figure 1) and higher formability [60–63] than the conventional quenched and tempered (Q&T) martensitic steel. In addition, the AMS possesses excellent toughness [53,64,65,74], high fatigue strength (especially high notch-fatigue strength) [75], and high delayed fracture strength [76]. So, the AMS is expected to be applied to not only press forming sheet products but also bar-forging products such as gear, screw, etc.

This paper introduces the microstructural and mechanical properties of various low and medium-carbon AMSs. In addition, the improvement mechanisms of the mechanical properties are detailed by relating to the matrix structure, the strain-induced transformation behavior of metastable retained austenite, and/or the martensite/austenite constituent (MA phase).

## 2. Two Kinds of Heat-Treatment Process for AMSs

The heat-treatment process of AMS is as simple as the Q&T process of the conventional martensitic steels. In addition, the process after austenitizing is conducted at relatively low temperature, compared with those of TBF, Q&P, and CFB steels [58–63,66–73]. This means that quenching in an oil bath can be used for the heat treatment, not a salt bath. Regarding alloying elements of the AMS, Si, and/or Al higher than 0.5 mass% are added to suppress the carbide precipitation and increase the volume fraction of carbon-enriched retained austenite [59–63]. In some cases, micro-alloying elements of Mn, Cr, Ni, Mo, B, etc. are added to increase the hardenability of the steels [62,65]. Furthermore, V, Ti, and/or Nb are added to refine the prior austenitic grain and to increase the strength through their carbide precipitates [27,59–63,67,68].

When the $M_f$ of the AMS is higher than room temperature, the IT process below $M_f$ [59–63] or DQ process [53,58–63,70–73] immediately after austenitizing and hot-rolling is conducted using an oil bath below 200 °C (Figure 2a). In the case of the DQ process, partitioning is added after the DQ process (named DQ-P process) [59,60,70–72]. The partitioning temperatures just below and above $M_f$ are recommended to minimize the increase in carbide precipitation and the decrease in retained austenite fraction [59,60]. Unlike this, Gao et al. propose the partitioning (tempering) temperature above $M_s$ [70–72]. Such IT and DQ-P processes (Figure 2a) are applied to Si/Al-Mn steels with low and medium carbon content [59–63] and with a medium manganese content of about 5% [65,66]. Regarding M-MMn steels, air cooling to room temperature is possible instead of quenching in oil or water bath due to the high hardenability [64,77–80].

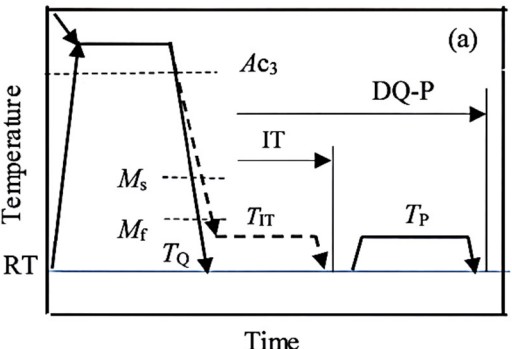
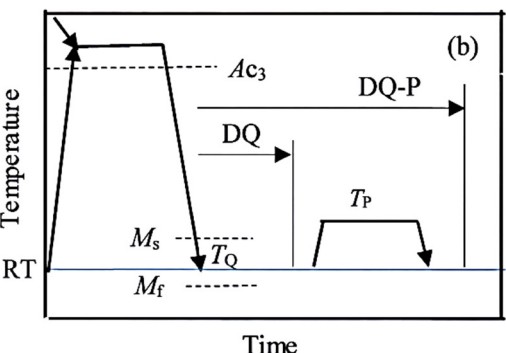

**Figure 2.** Heat treatment diagrams of (**a**) the isothermal transformation (IT) process below $M_f$ and direct quenching to room temperature (RT) followed by partitioning (DQ-P) process in a case of RT < $M_f$ [58–63] and (**b**) the DQ and DQ-P process in a case of RT > $M_f$ [67–69]. RT: room temperature, $T_{IT}$: isothermal transformation temperature, $T_Q$: quenching temperature, $T_P$: partitioning temperature.

On the other hand, when the $M_f$ of the AMS is lower than room temperature such as for M-MMn steel with a relatively high Mn content of about 10 mass %, the DQ or DQ-P process is carried out [67–69] (Figure 2b). For the DQ-P process, He et al. [67] adopt the partitioning (tempering) at 300 to 400 °C for 10 min in 0.2%C–10%Mn–2%Al–0.1%V steel.

To optimize the microstructure and mechanical properties of the AMS, ausforming at temperatures between $Ac_3$ and $M_s$ is carried out before the IT, DQ, or DQ-P process [68,81–83], in the same way as TBF [84] and CFB steels [49,85].

## 3. Microstructure and Retained Austenite Characteristics of AMSs

### 3.1. IT and DQ-P Processes (RT < $M_f$)

When 0.21%C–1.49%Si–1.50%Mn–1.0%Cr–0.05%Mn TM steel is subjected to the IT process at temperatures below $M_f$ (261 °C) and the DQ process, most of the austenite start to transform to soft coarse lath-martensite accompanied with auto-tempering [59–63] (Figures 3 and 4a). The other austenites are retained as filmy and blocky morphology in the lath-martensite structure matrix, and a part of retained austenite stays in the fine MA

phase [35,59–63]. It is noteworthy that fine lath/twin-martensite in the MA phase is very hard because of fine morphology, high dislocation density, and high carbon concentration. As mentioned in the previous section, partitioning after the DQ process rises the mechanical stability of the retained austenite due to carbon-enrichment. In addition, it plays an important role in the softening of the coarse lath-martensite and fine lath/twin-martensite and carbon enrichment of retained austenite, with a small increase in carbide fraction and a small decrease in retained austenite fraction [60], as well as the relaxation of transformation strain. Such the AMSs are corresponding to low and medium-carbon TM steels [32,59–63] and M-MMn steels with relatively low Mn content [65,66]. Note that carbides precipitate only in soft coarse lath-martensite (Figure 3c). De Cooman et al. show that the carbide is transition carbide or cementite [17].

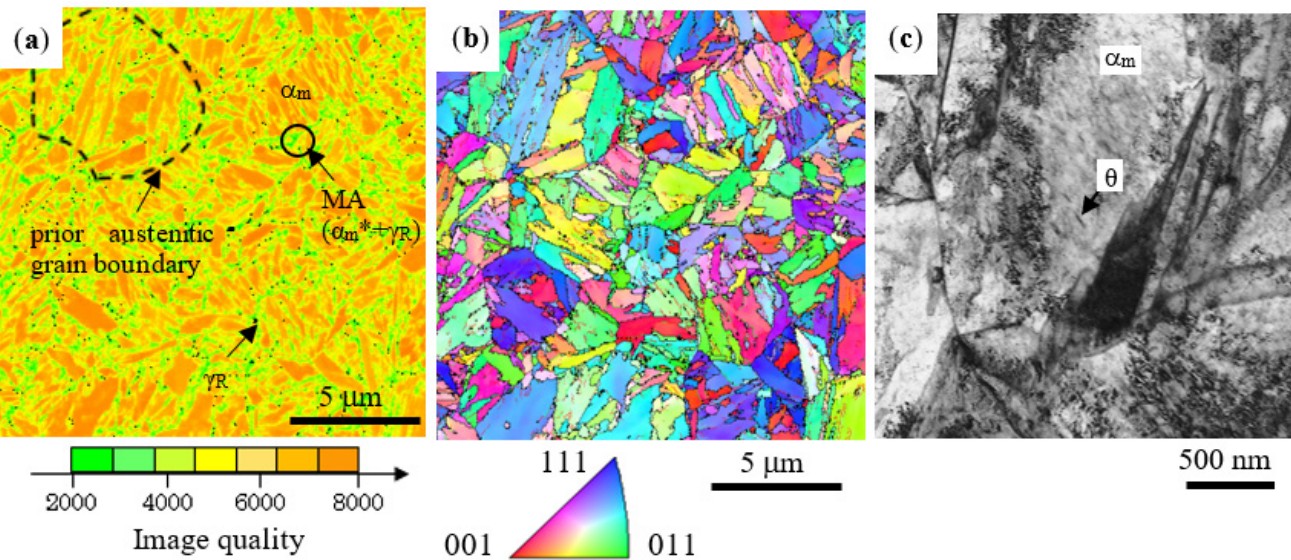

**Figure 3.** (**a**) Image quality distribution map and (**b**) inverse pole figure map of Fe-α (body centered cubic structure) phase and (**c**) TEM image in 0.21%C–1.49%Si–1.50%Mn–1.0%Cr–0.05%Mn TM steel subjected to the IT process at $T_p$ = 200 °C (< $M_f$) for 1000 s [62]. $α_m$: coarse lath-martensite, $α_m$*: fine lath/twin-martensite, $γ_R$: retained austenite (black dots), MA: martensite/austenite phase, θ: carbide in coarse lath-martensite. Reprinted with permission from ISIJ: ISIJ Int, Copyright 2021.

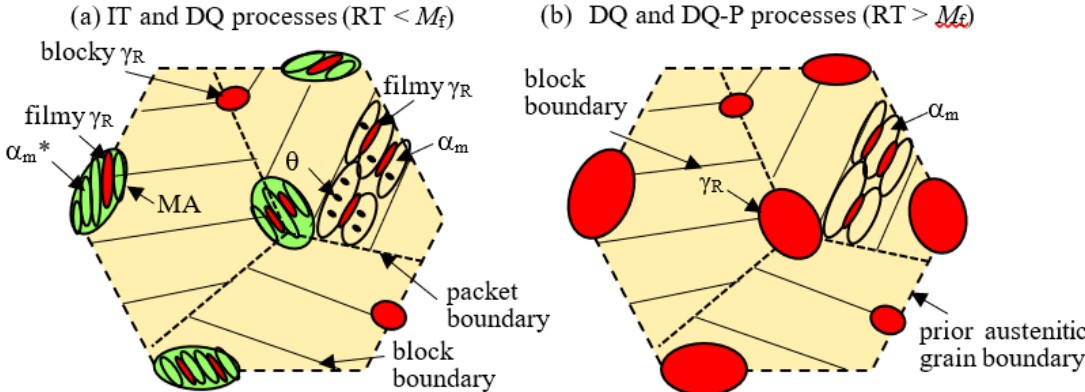

**Figure 4.** Illustration of typical microstructures of advanced martensitic steels (AMSs) subjected to (**a**) the IT and DQ processes in the case of RT < $M_f$ and (**b**) the DQ and DQ-P processes in the case of RT > $M_f$. RT: room temperature. $α_m$, $α_m$*, $γ_R$, θ and MA represent coarse lath-martensite, fine lath/twin-martensite, retained austenite, carbide and MA phase, respectively.

According to Sugimoto et al. [60,61,63], the retained austenite fraction increases with increasing IT temperature, with a constant carbon concentration, in an IT temperature range of 25 to 250 °C in 0.21%C–1.49%Si–1.50%Mn–1.0%Cr–0.05%Nb TM steel (Figure 5a), where the IT process at 25 °C is corresponding to the DQ process. It is noteworthy that the MA

phase fraction increases and the carbide fraction decreases with increasing IT temperature in the TM steel (Figure 5b). The carbide fraction is 1/4 to 1/2 times that of JIS-SCM420 Q&T steel. Additions of Cr and Mo hardly influence the retained austenite fraction (about 4 vol %) but increase the MA phase fraction and decrease the carbide fraction [59,62]. An increase in Mn content significantly increases the volume fraction of retained austenite in 0.2%C–1.5%Si–1.5%Mn TM steel [65,66]. The mechanical stability of the retained austenite defined by the following strain-induced transformation factor $k$ [6] is nearly constant in an IT temperature range below $M_f$ in the same way as the carbon concentration of the retained austenite (Figure 5b,c).

$$\log f\gamma = \log f\gamma_0 - k\,\varepsilon \tag{1}$$

where $f\gamma$ is the volume fraction of retained austenite after being subjected to a plastic strain $\varepsilon$ and $f\gamma_0$ is the initial volume fraction of retained austenite. The $k$-value is higher than that of TBF steel [29,30] owing to the lower carbon concentration of the retained austenite and higher flow stress. Mn addition in 0.2%C–1.5%Si–1.5%Mn steel increases the volume fraction and mechanical stability of retained austenite because Mn is austenite former element [65,66].

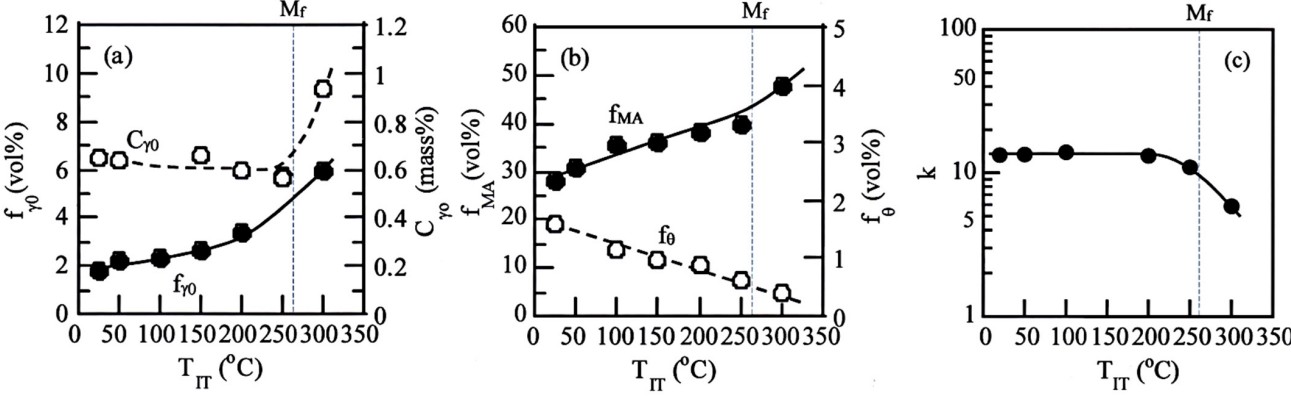

**Figure 5.** Variations in (**a**) initial volume fraction ($f\gamma_0$) and initial carbon concentration ($C\gamma_0$) of retained austenite, (**b**) volume fractions of MA phase ($f_{MA}$) and carbide ($f_\theta$), and (**c**) strain-induced transformation factor ($k$) as a function of IT temperature ($T_{IT}$) in 0.21%C–1.49%Si–1.50%Mn–1.0%Cr–0.05%Nb TM steel [61]. The IT process at 25 °C is corresponding to the DQ process. Reprinted with permission from AIST: AIST 2013, Copyright 2021.

### 3.2. DQ and DQ-P Processes (RT > $M_f$)

The DQ process at the temperatures above $M_f$ produces a simple duplex structure of soft coarse lath-martensite structure matrix and blocky and filmy retained austenite in 0.2%C–10%Mn–2%Al–0.1%V [67,68] (Figure 4b). The soft lath-martensite fraction ($f\alpha_m$) can be estimated by the following empirical equation proposed by Koistinen and Marburger [8,86].

$$f\alpha_m = 1 - \exp\{-1.1 \times 10^{-2}\,(M_s - T_Q)\} \tag{2}$$

where $T_Q$ is the quenching temperature. It is noteworthy that the retained austenite fraction is much higher than that of TM steel subjected to the IT process of Figure 2a [67,68]. Differing from the IT process (Figure 4a), the MA phase and carbide are hardly formed because the process is not cooled to temperatures below $M_f$, similar to TBF, Q&P, and CFB steels with a BF/M matrix structure [29–32,34,42–44,47]. In addition, such a microstructure without carbide in the soft lath-martensite resembles that of 0.23%C–2.3%Mn–1.5%Si–12.5%Cr–0.03%Ti–0.05%Nb martensitic stainless steel containing retained austenite [87].

According to Du et al. [69], in 0.24C–7.5Mn–1.1Si–0.1V M-MMn steel subjected to the DQ-P process, partitioning above $M_s$ softens the lath-martensite matrix structure due to the decreased carbon concentration and dislocation density, in the same way as TM steel subjected to the DQ-P process. On the contrary, the retained austenite is carbon-enriched.

Sub-cooling in liquid nitrogen decreases the retained austenite fraction. It is very important to know that the subsequent partitioning increases the volume fraction of retained austenite through the migration of austenite/martensite interface, with an increase in the carbon concentration. The AMS subjected to the DQ-P process is corresponding to M-MMn steels with $M_f$ lower than room temperature.

### 3.3. Ausforming

Ausforming at temperatures between $Ac_3$ and $M_s$ before the IT, DQ, and DQ-P process can not only reduce the $M_s$ due to the strengthening of austenite but also introduce deformation defects and elongate the microstructure [81], in the same way as TBF [84] and CFB [49] steels. If the ausforming is conducted under the conditions of relatively high temperature and large plastic strain, prior austenitic grain is refined, although dislocation density is decreased. According to He et al. [68], warm rolling at 600 °C before the DQ process is expected to avoid dynamic recrystallization and minimize dislocation recovery in 0.2%C–10%Mn–2%Al–0.1%V M-MMn steel. In this case, it is also essential to keep a finishing rolling temperature higher than the $M_s$ temperature (about 130 °C) to avoid martensitic transformation during the warm rolling process [68]. Hojo et al. [82,83] show that ausforming at 650 °C and subsequent IT process at 200 °C refines the microstructure and increases the retained austenite fraction and its stability in 0.23%C–1.5%Si–1.5%Mn–1.0%Cr–0.2%Mo–0.002%B TM steel.

## 4. Tensile Properties and Cold Formability of AMS Sheets

### 4.1. Tensile Properties

The IT process brings on the continuous yielding and low yield stress (or 0.2% offset proof stress) at room temperature in 0.21%C–1.49%Si–1.50%Mn–1.0%Cr–0.05%Nb TM steel (Figure 6). The main origin is due to a large amount of hard MA phase [61–63], which creates a preferential yielding zone in a matrix as fresh martensite second phase in dual-phase steel, as well as the strain-induced transformation of retained austenite [61–63,88].

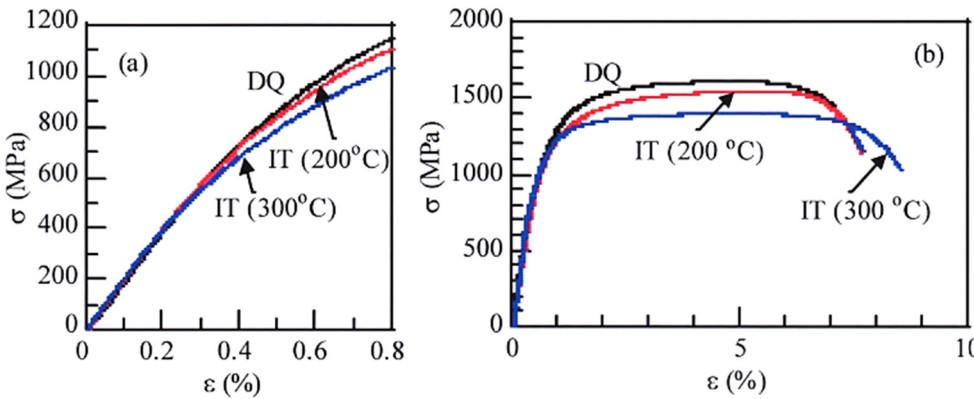

**Figure 6.** Typical engineering stress−strain ($\sigma$−$\varepsilon$) curves at room temperature in 0.21%C–1.49%Si–1.50%Mn–1.0%Cr–0.05%Nb TM steel subjected to the DQ process and the IT process at $T_p$ = 200 °C and 300 °C [63]. (**a**): initial stage, (**b**) overall stage [63]. Measured $M_s$ and $M_f$ of the steel are 406 °C and 261 °C, respectively.

When 0.21C–1.49Si–1.50Mn–1.0Cr–0.05Nb TM steel is subjected to the DQ process or the IT process at temperatures below 200 °C (< $M_f$), the tensile strength exceeds 1.5 GPa, although the tensile strength and yield stress slightly decrease with increasing IT temperature [60,61,63] (Figure 6). The uniform and total elongations and reduction of the area tend to slightly increase with increasing IT temperature. Partitioning after the DQ process increases the yield stress due to the relaxation of the internal residual stress, despite the softening of coarse and fine martensites and the coarsening of the carbides. In addition, the partitioning slightly decreases the uniform and total elongations [60]. Resultantly, the TM steel subjected to the DQ-P process exhibits a TS×TEl of about 11 GPa%. Torizuka and

Hanamura [53] show that 0.1%C–2%Si–5%Mn M-MMn steel subjected to the DQ process completes a TS×TEl of 30 GPa%, almost equal to those of D-MMn steel. He et al. [67] report that 0.2%C–10%Mn–2%Al–0.1%V M-MMn steel subjected to the DQ-P process ($T_p$ = 300 °C) achieves higher TS×TEl than press hardening steel and DP steel. Gao et al. find that slow cooling on the DQ process increases the total elongation with no change in tensile strength, compared to rapid cooling such as quenching in water [70].

### 4.2. Formabilities

The best combination of tensile strength and formabilities such as stretch-formability, stretch-flangeability, and bendability can be achieved by the IT process at $T_{IT}$ = 200 °C in 0.21%C–1.49%Si–1.50%Mn–1.0%Cr–0.05%Nb TM steel (Figure 7) [61]. This optimum IT temperature (200 °C) is equivalent to $M_f$—60 °C in the TM steel. All formabilities of the TM steel are superior to those of 22MnB5 Q&T steel and 0.082%C–0.88%Si–2.0%Mn DP steel. The increased stretch-formability may be caused by the TRIP effect of metastable retained austenite. Meanwhile, the high stretch-flangeability is brought from a small degree of damage to the punched hole-surface with a long shear section and a small number of tiny cracks or voids, resulting from the plastic relaxation by the strain-induced transformation of retained austenite. Such small punching damage leads to difficult crack propagation and void growth on hole-expanding [60,61]. Good bendability is considered to be caused by a high localized ductility. Partitioning after the DQ process further increases these formabilities, but the formabilities are lower than those of the IT process at 200 °C [60].

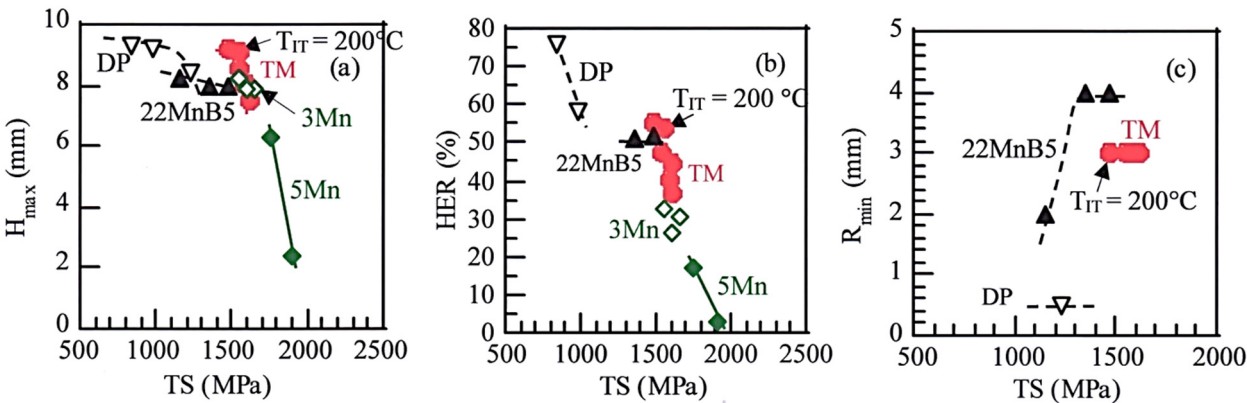

**Figure 7.** Relationships between (**a**) maximum stretch height ($H_{max}$), (**b**) hole expansion ratio (HER) and (**c**) minimum bending radius ($R_{min}$), and tensile strength (TS) in 0.21%C–1.49%Si–1.50%Mn–1.0%Cr–0.05%Nb TM steel subjected to the DQ process or the IT process at $T_{IT}$ = 100 to 250 °C for 1000 s (●). DP (▽): 0.082%C–0.88%Si–2.0%Mn ferrite–martensite dual-phase steel, DIN-22MnB5 (▲): 0.23%C–0.19%Si–1.29%Mn–0.21%Cr–0.003%B Q&T steel [61], 3Mn (◇): 0.2%C–1.5%Si–3%Mn M-MMn steel subjected to the DQ process and the IT process at $T_{IT}$ = 200 °C, 5Mn (◆): 0.2%C–1.5%Si–5%Mn M-MMn steel subjected to the DQ process and the IT process at 100 °C [66]. This figure was reproduced from Refs. [61,66]. Ref. [61] is reprinted with permission from AIST: AIST 2013, Copyright 2021.

Sugimoto and Tanino [66] show that 0.2%C-1.5%Si-5%Mn M-MMn steel subjected to the IT process at 100 °C ($<M_f$) achieves a maximum stretch height ($H_{max}$) of 6.3 mm and a hole expanding ratio (HER) of 16.7% at maximum (Figure 7). However, 0.2%C-1.5%Si-3%Mn M-MMn steel with low Mn content exhibits higher $H_{max}$ and HER, although the HER is lower than that of 22MnB5 Q&T steel [89]. Sugimoto et al. propose that a large amount of retained austenite transformed at an early stage in the M-MMn steels contradicts the increase in the stretch-formability and stretch-flangeability through the easy void initiation and growth [66]. Up to now, there is no data of bendability in the M-MMn steel.

## 5. Mechanical Properties of Heat-Treated AMS Plates and Bars

### 5.1. Impact Toughness and Fracture Toughness

According to Kobayashi et al. [74], 0.2%C-1.5%Si-1.5%Mn-(0–1.0)%Cr-(0–0.2)%Mo TM steels subjected to the DQ-P process ($T_p$ = 300 °C) possess as high Charpy V-notch impact value ($E_v$) at room temperature as TBF steels with the same chemical composition (Figure 8a). The $E_v$s and the product of TS and $E_v$ (TS×$E_v$) of 0.2%C-1.5%Si-(3, 5)%Mn M-MMn steels subjected to the DQ or the IT process decrease compared with those of TM steels, especially in 5%Mn steel [65]. Ductile-brittle transition temperatures (DBTTs) of the TM steels are far lower than those of JIS-SCM420 Q&T and TBF steels [74,90] (Figure 8b). The DBTTs of the M-MMn steels are almost the same as those of 0.2%C-1.5%Si-1.5%Mn-(0–1.0)%Cr-(0–0.2)%Mn TBF steels but are far higher than the TM steels [65].

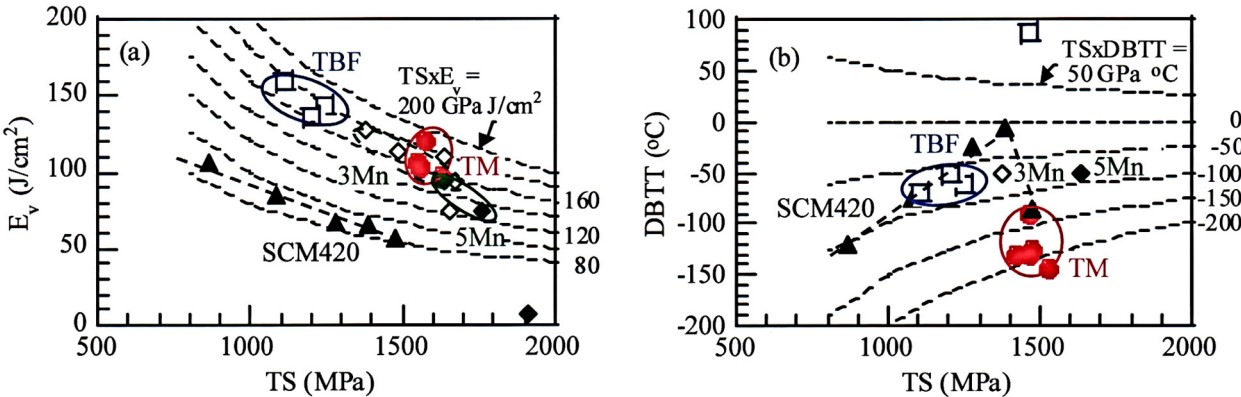

**Figure 8.** (**a**) Relationships between (**a**) Charpy V-notch impact value ($E_v$) at 25 °C and (**b**) ductile–brittle fracture transition temperature (DBTT) and tensile strength (TS) in 0.2%C–1.5%Si–1.5%Mn–(0–1.0)%Cr–(0–0.2)%Mn TBF ($T_{IT}$ = 400 °C, □) and TM (DQ-P, $T_p$ = 300 °C, ●) steels, JIS-SCM420 Q&T steel ($T_T$ = 200 °C to 600 °C, ▲) and 0.2%C–1.5%Si–3%Mn (◇) and 0.2%C–1.5%Si–5%Mn (◆) M-MMn steels. This figure was reproduced from refs. [65,74]. Ref. [74] was reprinted with permission from Springer Nature: Metall. Mater. Trans. A, Copyright 2021.

Figure 9 illustrates (a) the ductile fracture (void initiation, growth and void-coalescence) and (b) brittle fracture (cleavage crack initiation and propagation) behavior of the TM steel appeared on impact tests [74,90]. In the ductile fracture, most of the voids originate at the interface of the large MA phase and the matrix structure. In this case, retained austenite suppresses the void initiation through the plastic relaxation by the strain-induced martensite transformation. As the deformation progresses further, these voids grow and coalesce (Figure 9a). Therefore, (i) a large amount of MA phase and (ii) the plastic relaxation mainly contribute to the high $E_v$s of the TM steels. Gao et al. show that 0.20%C–2.0%Mn–1.5%Si–0.5%Cr–0.28%Mo and 0.4%C–2.0%Mn–1.7%Si–0.4%Cr TM steels subjected to the DQ-P process ($T_p$ = 280 °C > $M_s$) exhibit higher $E_v$s than TBF and CFB steels, although the $E_v$s of these steels are lower than that of two-step Q&P steels with the same chemical composition [71,72].

In the brittle fracture, the strain-induced transformation of retained austenite also suppresses the cleavage crack initiation, and the MA phase retards the crack growth (Figure 9b). Resultantly, the retained austenite and MA phase play a role in lowering the DBTT of the TM steel [74]. On the other hand, high DBTTs of 0.2%C–1.5%Si–(3,5)%Mn M-MMn steels are mainly associated with high solute Mn concentration in the matrix, although high Mn addition increases the volume fraction and mechanical stability of retained austenite and decreases the unit path of quasi-cleavage crack ($L_c$ in Figure 9b) [65].

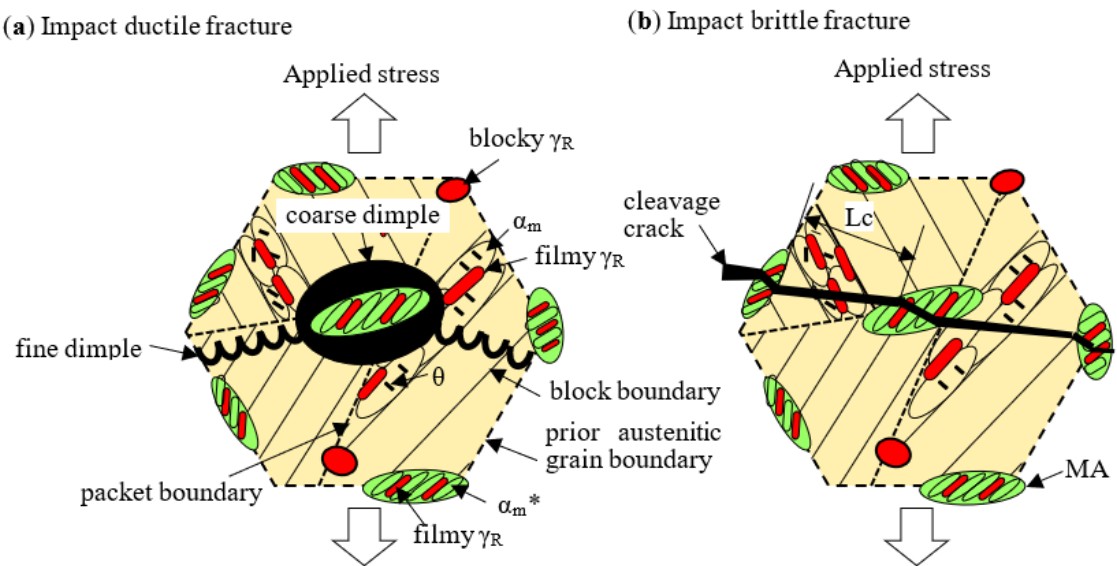

**Figure 9.** Illustrations showing (**a**) ductile fracture and (**b**) brittle fracture of TM steel appeared on impact tests [74]. *Lc*: Quasi-cleavage length affected by the MA phase located on prior austenitic, packet, and block boundaries. $\alpha_m$, $\alpha_m^*$, $\gamma_R$, $\theta$, and MA represent coarse lath-martensite, fine lath/twin-martensite, retained austenite, carbide, and MA phase, respectively. Reprinted with permission from Springer Nature: Metall. Mater. Trans. A, Copyright 2021.

Kobayashi et al. find that 0.2%C–1.5%Si–1.5%Mn–1.0%Cr–0.05%Nb TM steel subjected to the DQ-P process ($T_p$ = 200 to 450 °C) [91] and the IT process ($T_{IT}$ = 200 °C and 250 °C, $<M_f$) [92] shows extremely high fracture toughness ($K_{IC}$ = 132–163 MPa m$^{1/2}$) (Figure 10). The fracture toughness is two times that ($K_{IC}$ = 57–63 MPa m$^{1/2}$) of JIS-SCM420 Q&T steel tempered at $T_T$ = 200 to 400 °C and is the same degree as that of 18Ni maraging steel. It is considered that the superior fracture toughness is essentially due to (i) a large amount of MA phase and (ii) plastic relaxation by the strain-induced transformation of the retained austenite in the same way as the above impact toughness, which suppresses crack formation, growth, and coalescence as well as cleavage fracture at the pre-crack tip. Wu et al. [93] report that fracture toughness of 0.20%C–1.42%Si–1.87%Mn steel subjected to the DQ-P process ($T_p$ = 300 °C) is only $K_{IC}$ = 54.5 MPa m$^{1/2}$. The low fracture toughness significantly differs from the result of Figure 10. This result should be discussed in the future.

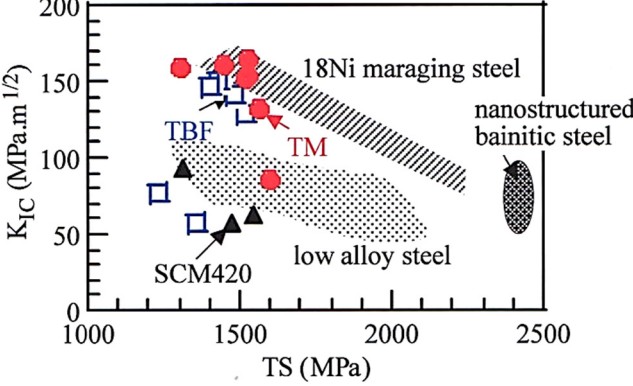

**Figure 10.** Relationship between fracture toughness ($K_{IC}$) and tensile strength (TS) [92] in 0.2%C–1.5%Si–1.5%Mn–(0–1.0)%Cr–(0–0.2)%Mn TBF (□) and TM (DQ-P, $T_p$ = 200 °C to 450 °C, ●) steels and JIS-SCM420 Q&T steel ($T_T$ = 200 °C to 400 °C, ▲), compared with low alloy Q&T, 18Ni maraging and nanostructured bainitic steels [92,94]. This figure is reproduced based on Ref. [92]. Reprinted with permission from ISIJ: ISIJ Int, Copyright 2021.

### 5.2. Fatigue Property

0.2%C–1.5%Si–1.5%Mn–1.0%Cr–0.2%Mo TM steel subjected to the DQ-P ($T_p$ = 200 °C) process exhibits large fatigue hardening despite a tensile strength over 1.5 GPa [95] (Figure 11a), in the same way as TBF [96], CFB [97], and high alloy TRIP (16%Cr–7%Mn–8%Ni and 16%Cr–6%Mn–6%Ni) [98] steels. Conversely, conventional martensitic steel such as JIS-SCM420 Q&T steel exhibits fatigue softening.

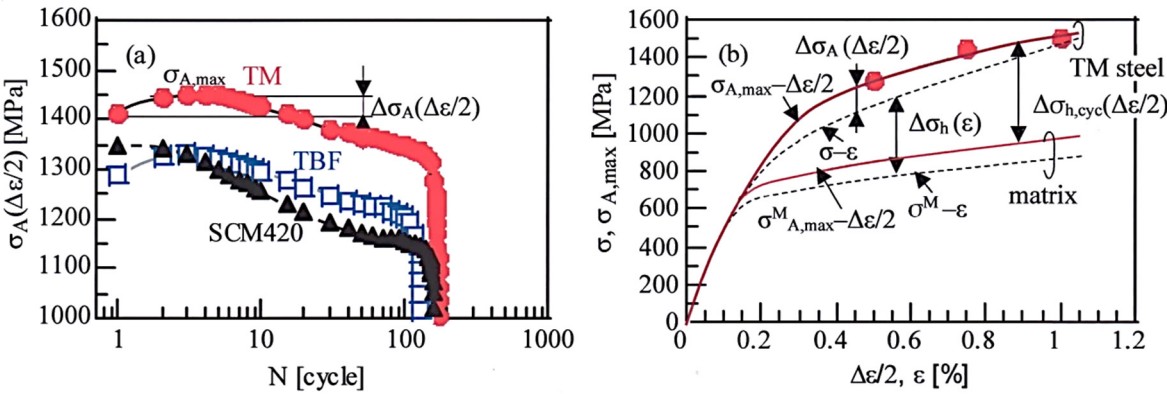

**Figure 11.** (**a**) Variations in stress amplitude ($\sigma_A(\Delta\varepsilon/2)$) as a function of the number of cycles ($N$) under total strain amplitude of $\Delta\varepsilon$ = 1.5% for 0.2%C–1.5%Si–1.5%Mn–1.0%Cr–0.2%Mo TM subjected to the IT process at $T_{IT}$ = 200 °C and TBF steel ($T_{IT}$ = 350 °C), and JIS-SCM420 Q&T steel ($T_T$ = 200 °C). (**b**) Monotonic stress–strain ($\sigma$-$\varepsilon$: dotted lines) and maximum cyclic stress amplitude–strain amplitude ($\sigma_{A,max}$-$\Delta\varepsilon/2$: solid red lines) curves for TM steel, as well as the definition of $\Delta\sigma_h(\Delta\varepsilon/2)$, $\Delta\sigma_{h,cyc}(\Delta\varepsilon/2)$, and $\Delta\sigma_A(\Delta\varepsilon/2)$, in which $\sigma^M$-$\varepsilon$ and $\sigma^M_{A,max}$-$\Delta\varepsilon/2$ are the monotonic and cyclic curves of the assumed matrix structure in the TM steel, respectively [95]. Reprinted with permission from Springer Nature: Metall. Mater. Trans. A, Copyright 2021.

In general, true cyclic hardening increment $\Delta\sigma_{h,cyc}(\Delta\varepsilon/2)$ of the TM steel (see Figure 11b) is obtained by [95]

$$\Delta\sigma_{h,cyc}(\Delta\varepsilon/2) = \sigma_{A,max}(\Delta\varepsilon/2) - \sigma^M_{A,max}(\Delta\varepsilon/2) = \Delta\sigma_{i,cyc}(\Delta\varepsilon/2) + \Delta\sigma_{t,cyc}(\Delta\varepsilon/2) \\ + \Delta\sigma_{f,cyc}(\Delta\varepsilon/2) \tag{3}$$

where $\sigma_{A,max}(\Delta\varepsilon/2)$ and $\sigma^M_{A,max}(\Delta\varepsilon/2)$ are the maximum cyclic stress amplitudes of TM steel and its matrix, respectively. $\Delta\varepsilon$ is total strain amplitude. Additionally, $\Delta\sigma_{i,cyc}(\Delta\varepsilon/2)$, $\Delta\sigma_{t,cyc}(\Delta\varepsilon/2)$, and $\Delta\sigma_{f,cyc}(\Delta\varepsilon/2)$ represent "the long-range internal stress hardening", "the strain-induced transformation hardening", and "the forest dislocation hardening of the matrix" upon cyclic deformation, respectively, which can be formulated by

$$\Delta\sigma_{i,cyc}(\Delta\varepsilon/2) = \{(7-5\nu)\mu/5(1-\nu)\} f\cdot\varepsilon_p{}^\mu \tag{4}$$

$$\Delta\sigma_{t,cyc}(\Delta\varepsilon/2) = g(\Delta f\,\alpha_m) \tag{5}$$

$$\Delta\sigma_{f,cyc}(\Delta\varepsilon/2) = \zeta\mu(b\cdot f\cdot\varepsilon/2r)^{1/2} \tag{6}$$

where $\nu$ is the Poisson's ratio, $\mu$ is the shear modulus, $\varepsilon_p{}^\mu$ is the eigenstrain, $f$ is the volume fraction of the second phase, $g(\Delta f\,\alpha_m)$ is a function of the strain-induced martensite fraction, $\zeta$ is a material constant, $b$ is the Burgers vector, and $r$ is the particle radius of the second phase. From the estimation of each cyclic hardening of Equations (4)–(6), Sugimoto et al. [95] propose that the fatigue hardening of TM steel is mainly associated with the compressive internal stress resulting from a difference in the flow stress (or plastic strain) between the matrix and the MA phase, with a small contribution from the strain-induced transformation and forest dislocation hardenings. The above theory is also applied to the fatigue hardening behavior of TPF [99] and TBF [95] steels.

The IT process enhances the smooth- and notch-fatigue limits ($\sigma_w$ and $\sigma_{wn}$) of (0.1–0.4)% C–1.5%Si–1.5%Mn TM steels and resultantly reduces the notch-sensitivity factor in a Vickers hardness range between HV350 and HV600, compared to the conventional JIS-SCM420, SCM435, and SCM440 Q&T steels (Figure 12a) [75]. However, the notch-fatigue limits are lower than those of TBF steels [100]. In addition, the notch-sensitivity factor is higher than that of 0.2%C–1.5%Si–1.5%Mn–(0–1.0)%Cr–(0–0.2)%Mo–(0–1.5)%Ni TBF steels in a hardness range below 400HV. The above-mentioned notch-sensitivity factor for fatigue limit ($q$) is defined by the following equation [101].

$$q = (K_f - 1)/(K_t - 1) \tag{7}$$

where $K_f$ and $K_t$ are the fatigue notch factor (=$\sigma_w/\sigma_{wn}$) and the elastic stress concentration coefficient ($K_t$ = 1.7 in Ref. [75]), respectively.

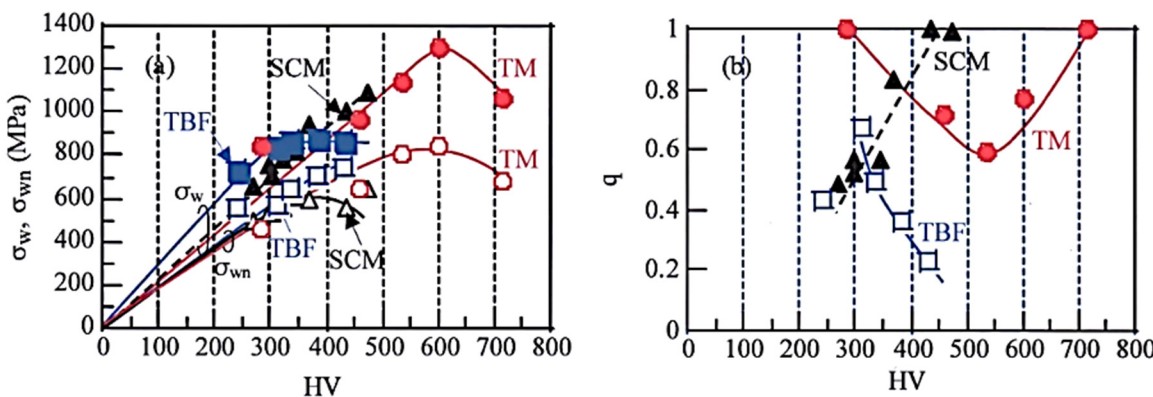

**Figure 12.** Variations in (**a**) fatigue limits ($\sigma_w$, $\sigma_{wn}$) of smooth and notched specimens and (**b**) notch-sensitivity ($q$) as a function of Vickers hardness (HV) in (0.1–0.6)%C–1.5%Si–1.5%Mn TM steels subjected to DQ-P process ($T_p$ = 250 °C, ●○), 0.2%C–1.5%Si–1.5%Mn–(0–1.0)%Cr–(0–0.2)%Mo–(0–1.5)%Ni TBF steels ($T_{IT}$ = 400 °C, ■□) and JIS-SCM420, 435 and 440 Q&T steels ($T_T$ = 200–600 °C, ▲△). This figure was reproduced from Refs. [75,100]. Refs. [75,100] are reprinted with permission from ISIJ: ISIJ Int, Copyright 2021 and from Springer nature: Metall. Mater. Trans. A, Copyright 2021, respectively.

According to Knott [102], the plastic zone size ($d_Y$) at a small fatigue crack tip can be estimated using the following equation,

$$d_Y = K^2/(3\pi YS^2) \tag{8}$$

where $K$ is the stress intensity factor defined by $K = \sigma(\pi c)^{1/2}$ ($\sigma$ is the applied stress and $c$ is the crack length) and YS is the yield stress of the material. The plastic zone size is estimated to be about 4.0 μm in 0.4%C–1.5%Si–1.5%Mn TM steel if the fatigue crack length at the first stage is 2c = 30 μm equivalent to the prior austenitic grain size and the applied stress is the maximum stress corresponding to the fatigue limit [75].

As shown in Figure 13, the plastic zone always includes some retained austenite particles in the MA phase because the inter-particle path of the MA phase is 0.5–2.0 μm, as well as a matrix structure. Based on this result, Kobayashi et al. [75] propose that the high fatigue limits and low notch-sensitivity of (0.1–0.4)%C–1.5%Si–1.5%Mn TM steels are principally associated with (i) the plastic relaxation of localized stress concentration as a result of the strain-induced transformation of 3–5 vol % metastable retained austenite and (ii) a large amount of finely dispersed MA phase along prior austenitic, packet, and block boundaries, as well as (iii) a small amount of carbide only in the coarse lath-martensite structure, which contributes to difficult fatigue crack initiation and/or propagation.

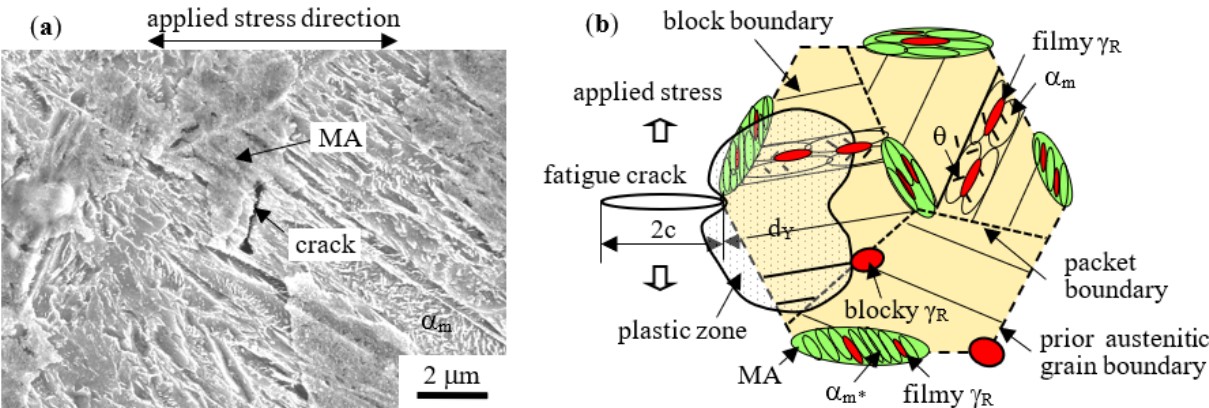

**Figure 13.** (**a**) SEM image of the initial crack formed on the notched surface of 0.4%C–1.5%Si–1.5%Mn TM steel (DQ-P, $T_p$ = 250 °C) failure at $N_f$ = 5.0 × 10$^4$ cycles [75]. (**b**) Illustration of the plastic zone size ($d_Y$) at the crack tip and distribution of MA phases in the TM steel [75]. $\alpha_m$, $\alpha_m^*$, $\gamma_R$, and θ represent coarse lath-martensite and fine lath/twin-martensite in the MA phase, retained austenite and carbide, respectively. Reprinted with permission from ISIJ: ISIJ Int, Copyright 2021.

A similar mechanism is also reported by Tomita et al. using 0.6%C–1.5%Si–0.8%Mn Q&P steel with BF/M duplex phase [103] and Huo and Gao using 0.24%C–1.75%Si–1.79%Mn–1.74%Ni–1.06%Cr–0.32%Mo–0.13%V steel subjected to the DQ-P process ($T_p$ = 150 to 550 °C) [104]. Zao et al. report that Nb addition of 0.042% in 0.21%C–1.74%Si–2.20%Mn–0.62%Cr TM steel subjected to the DQ-P process ($T_p$ = 280 °C) increases a very high cycle fatigue strength because of the cumulative effect of strengthening associated with grain refinement and precipitation strengthening mechanisms [105].

### 5.3. Delayed Fracture Strength

It is well-known that metastable retained austenite can absorb a large amount of solute hydrogen in TBF, Q&P, CFB, and D-MMn steels [57,106–110]. This results in a high delayed fracture strength in these steels because hydrogen concentration on the prior austenitic grain boundary is lowered. In 0.2%C–1.5%Si–1.5%Mn–(0–1.0)%Cr–(0–0.2)%Mo–(0–1.5%%Ni–0.05%Nb TM steels, the DQ-P process ($T_p$ = 250 and 350 °C) increases the delayed fracture strength (maximum fracture strength enduring for 5 h measured by four-point bending tests). In addition, the DQ-P process decreases hydrogen embrittlement susceptibility (HES), which is defined by the following equation in the TM steels [76,108] (Figure 14a).

$$\text{HES} = (\varepsilon_0 - \varepsilon_1)/\varepsilon_0 \times 100\% \tag{9}$$

where $\varepsilon_0$ and $\varepsilon_1$ represent the total elongation of steel before and after hydrogen charging, respectively. The HES values are comparable with those of TBF steels with the same chemical composition and far lower than those of 0.40%C–0.16%Si–1.44%Mn Q&T steel [106–108]. The HES of the TM steels can be improved by the addition of micro-alloying elements, especially 1.0%Cr addition (corresponding to steel D of TM steel in Figure 14) [63]. In the 1.0%Cr TM steels, the metastable retained austenite plays a role in trapping the diffusible hydrogen, similar to the third-generation AHSSs (Type A) [63,76] (Figure 14b). Hojo et al. [76] propose that the high hydrogen embrittlement resistance is caused by that (i) most of the solute hydrogen is trapped in the retained austenite with high mechanical stability, and resultantly, (ii) the initiation and propagation of voids and cracks at prior austenite grain, packet, block, and lath boundaries are suppressed.

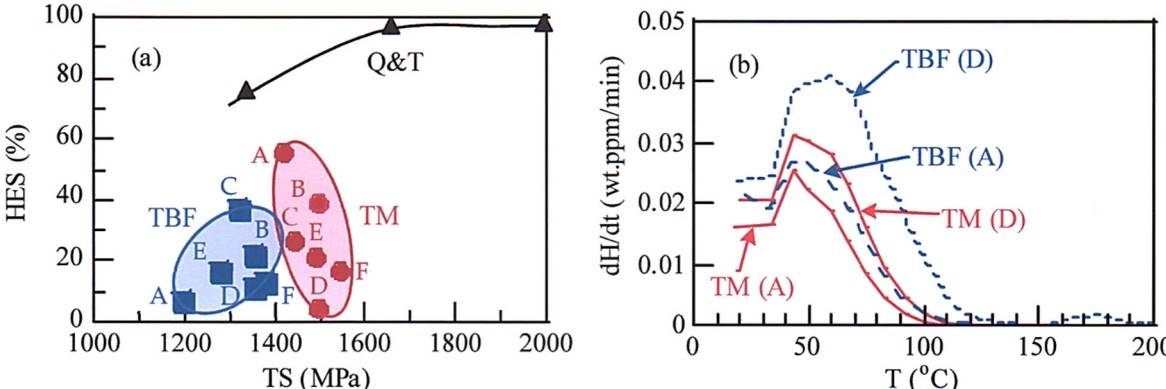

**Figure 14.** (**a**) Relationship between hydrogen embrittlement susceptibility (HES) and tensile strength (TS) [63] and (**b**) hydrogen evolution curves of 0.2%C–1.5%Si–1.5%Mn–(0–1.0)%Cr–(0–0.2)%Mo–0.05%Nb TM steels subjected to DQ-P process ($T_p$ = 250 °C, 300 °C, ●) and TBF steels ($T_{IT}$ = 350 °C, ■) and 0.40%C–0.16%Si–1.44%Mn Q&T steel ($T_T$ = 200 °C to 400 °C, ▲) [63]. A: 0.20%C–1.50%Si–1.50%Mn, B: 0.20%C–1.52%Si–1.50%Mn–0.05%Nb, C: 0.21%C–1.49%Si–1.50%Mn–0.5%Cr–0.05%Nb, D: 0.20%C–1.49%Si–1.50%Mn–1.0%Cr–0.05%Nb, E: 0.18%C–1.48%Si–1.49%Mn–1.0%Cr–0.2%Mo–0.05%Nb, F: 0.21%C–1.49%Si–1.49%Mn–1.52%Ni–1.0%Cr–0.2%Mo–0.05%Nb.

Many researchers are investigating the hydrogen embrittlement of D-MMn steels [57,111,112]. Unfortunately, there is not any research on the delayed fracture strength of M-MMn steel up to now.

### 5.4. Wear Property and Overall Performance of Mechanical Properties

There are many kinds of research on the wear properties of TBF, Q&P, and CFB steels with BF/M matrix structure [97,113,114]. However, the research on the wear property of TM and M-MMn steels is hardly presented. According to De Oliveira et al. [113], the two-step Q&P process improves the wear performance compared to the one-step Q&P process ($T_{IT} > M_s$) in 0.29%C–1.35%Si–1.96%Mn–0.1%Cr–0.03%Mo–0.04%Nb steel. Feng et al. [97] show using 0.24%C–1.44%Si–1.76%Mn–0.75%Ni–1.76&Cr–0.39%Mo–0.65%Al rail steel that the DQ process (air cooling) after austenitizing enhances the wear resistance compared to the IT process above $M_s$. They propose that the strain-induced martensite transformation of retained austenite in the steel can be linked to the retardation of crack initiation and propagation, as well as martensite formed on air cooling. Wang et al. [114] show that partitioning and cryogenic treatment after the IT process ($T_{IT}$ = 360 °C and 400 °C > $M_s$) followed by slow cooling is effective to increase the wear property in 0.22%C–2.0%Mn–1.0%Si–0.8%Cr–0.8%(Mo+Ni) BF/M rail steel. This is mainly associated with a decrease in the volume fraction of retained austenite or an increase in the hardness because surface hardness is the main factor controlling the abrasive wear [115]. From the above results of TBF, Q&P, and CFB rail steels, we can expect that TM and M-MMn steels possess good wear resistance because the hardness is higher than the TBF, Q&P, and CFB steels, although the retained austenite fraction is lower.

When various mechanical properties without wear property of 0.2%C–1.5%Si–1.5Mn TM steel are compared to those of TBF steel with the same composition and DIN-22MnB5 Q&T steel, the TM steel is superior to that of Q&T steel (Figure 15) [116,117]. The mechanical properties of M-MMn steel are not ready yet. Ausforming at temperatures between $Ac_3$ and $M_s$ is expected to enhance the whole mechanical properties. However, only the effects of ausforming conditions on the tensile strength, total elongation, and impact toughness are reported up to now [118,119].

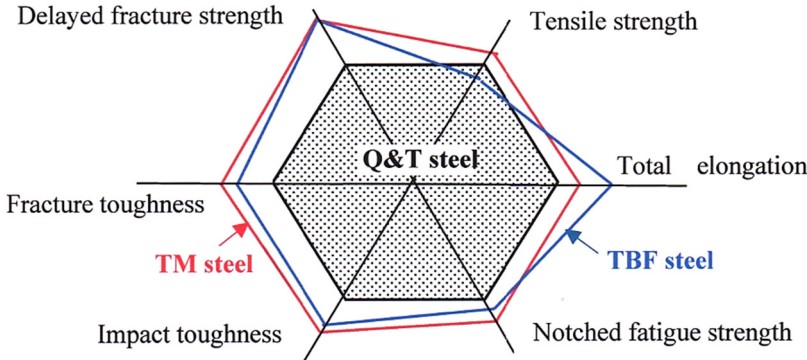

**Figure 15.** Comparison of various mechanical properties of 0.2%C–1.5%Si–1.5Mn TBF and TM steels and DIN-22MnB5 Q&T steel. This figure was modified based on Ref. [117].

## 6. Hot- and Warm-Workings to Produce Automotive AMS Parts

Hot-workings such as hot-stamping and hot-forging refine the microstructure and increase the retained austenite fraction in AMSs, in the same way as ausforming. Resultantly, the hot-workings enhance strength and other mechanical properties. In the following, the tensile and mechanical properties of hot-stamped and hot-forged AMSs are introduced.

### 6.1. Hot- and Warm-Stamping

In the conventional hot-stamping, the steel blank is completely austenitized at temperatures above $Ac_3$ before stamping, followed by die-quenching to 150 to 250 °C [120,121]. The hot-stamping process considerably reduces the spring back of the products, differing from the cold-stamping process [120]. As the process suppresses the hydrogen embrittlement due to lowering the residual stress, hot-stamping steels such as DIN-22MnB5 and 30MnB5 steels are applied to automotive center pillars with the tensile strength of 1.5 to 1.8 GPa [120,122]. The matrix structure after die-quenching is almost martensite.

Recently, warm-stamping at temperatures above $M_s$ are developed to improve productivity and reduce manufacturing costs [122–125] (Figure 16). In the warm-stamping, the blank is austenitized at temperatures above $Ac_3$, in the same way as the conventional hot-stamping process. After that, the austenitized blanks are pre-cooled to any temperatures during $Ac_3$ and $M_s$ before stamping. By this warm-stamping, the blank is strain-hardened along with the improvement of the drawability and formability. In addition, productivity can also be increased by shortening the cooling cycle time and decreasing the oxidation [122]. At present, the warm-stamping at 450 to 500 °C is applied to the B-pillar of (0.08–0.2)%C–(4.0–7.0)%Mn D-MMn steels, not M-MMn steel [125]. After the warm-stamping, the tensile strength is about 1.4 GPa with total elongation of 11.8%.

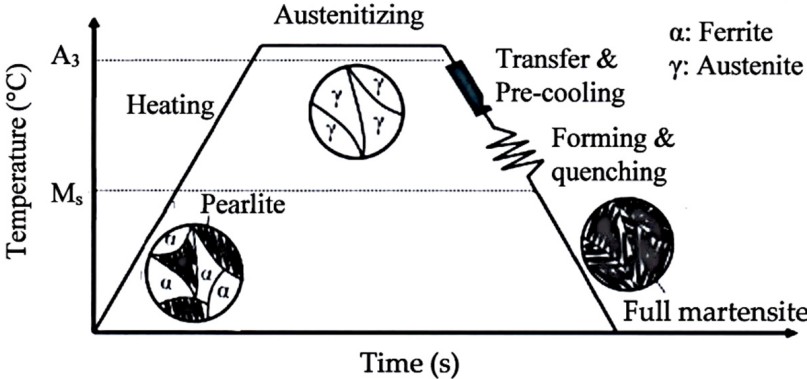

**Figure 16.** Schematic illustration of hot-stamping process at low-temperature and warm temperature [122].

As another latest advance, the combined hot-stamping and two-step Q&P process is proposed [57,126]. In 0.28%C–1.69%Si–1.10%Mn–0.99%Cr–0.029%Nb steel, in which the two-step Q&P process brings on the matrix structure containing martensite. This combined process is also applied to the TM steel subjected to the IT and DQ-P processes [127].

*6.2. Hot- and Warm-Forging*

Up to now, the conventional forging steels such as precipitation hardening ferrite/pearlite (PHFP) steels, micro-alloyed PHFP (PHFP-M) steels [128,129], bainitic steels [130,131] and Q&T martensitic steels [132,133] are applied to the automotive engine, powertrain and chassis for weight reduction. For further weight reduction of their parts, 1.5 to 2.0 GPa grade hot-forging AMSs such as TM [118,119,129] and M-MMn [78–80] steels subjected to the IT, DQ, and DQ-P processes have been recently developed, as well as TBF [80,134–136], Q&P [137], and CFB [138,139] forging steels with the BF/M matrix. In general, this prospective AMSs contains Si and/or Al higher than 1.0 mass % to suppress the carbide formation and promote a predominant formation of carbon-enriched retained austenite [118,119,140]. Micro-alloying of Mn, Cr, Ni, Mo, B, etc. is fundamental to increase the hardenability. In addition, additions of V, Ti, and/or Nb are required for refining the prior austenitic grain and precipitation hardening.

According to Kobayashi et al. [118], hot-forging at 900 °C (>$Ac_3$) with a reduction strain of 40 to 60% (one pass, strain rate: 50%/s) followed by the DQ-P process ($T_p$ = 200 to 350 °C) produces a uniform and fine microstructure in 0.4C–1.5Si–1.5Mn–0.05Nb–0.002B TM steel, without polygonal ferrite. In addition, the hot-forging increases the volume fractions of retained austenite and MA phase, with the decreased carbide fraction. Resultantly, the hot-forging and subsequent DQ-P process bring on an excellent combination of yield stress and Charpy V-notch impact value (YS = 1400 to 1561 MPa and $E_v$ of 35 to 44 J/cm$^2$) (Figure 17) [117,118]. The balance is far higher than the conventional Q&T martensitic, bainitic, PHFP, and PHFP-M steels [141–146], and it is nearly the same as those of TBF, Q&P, and CFB steels with a BF/M matrix structure, although the balance is slightly inferior to that of TBF steels with the same chemistry. According to Kobayashi et al. [118,119], the increased balance is associated with (i) refined duplex phase structure (soft coarse lath-martensite and MA phase) and (ii) increased volume fractions of retained austenite and MA phase. Note that the balance (YS×$E_v$) value of D-MMn steel is lower, although the steel possesses an extremely high TS×TEl value.

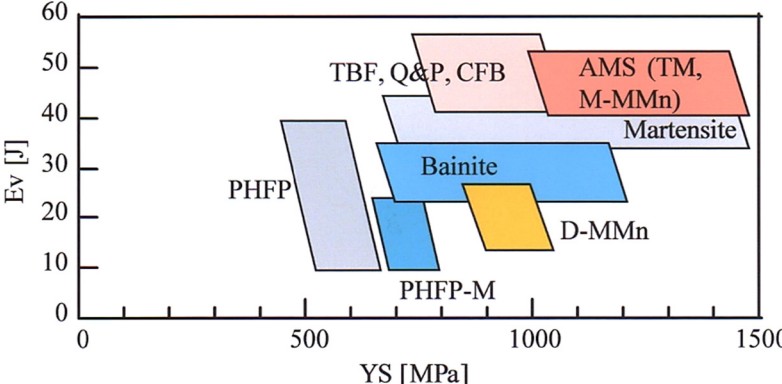

**Figure 17.** Relationship between Charpy V-notch impact value ($E_v$) and yield stress (YS) at room temperature in steel groups [77,117,141–146]. PHFP, PHFP-M, TBF, Q&P, CFB, D-MMn, AMS, TM and M-MMn are precipitation-hardening ferritic–pearlitic, micro-alloyed PHFP, TRIP-aided bainitic ferrite, quenching and partitioning, carbide-free bainitic, duplex-type medium Mn, advanced martensitic, TRIP-aided martensitic, and martensite-type medium Mn steels, respectively. This figure is reproduced based on Ref. [117].

The cooling rate just after hot-forging is a very important factor controlling the microstructure and mechanical properties of AMSs. Kobayashi et al. [119] show an important result that a low cooling rate (1.2 °C/s) to room temperature after hot-forging increases the $E_v$ and lowers the DBTT in 0.21%C–1.49%Si–1.49%Mn–1.0%Cr–0.20%Mo–1.50%Ni–0.05%Nb TM steel. In addition, they show that the improved impact toughness is mainly caused by the increased volume fraction and carbon concentration of retained austenite and the decreased carbide fraction as well as the finely dispersed MA phase. Gramlich et al. [80] show that air cooling to room temperature after hot-forging produces a mixture of martensite with a small amount of carbide and retained austenite and results in extremely high strain-hardening in 0.18%C–0.52%Si–3.98%Mn–0.54%Al–0.11%Cr–0.20%Mo–0.10%Ni–0.11%V–0.036%Nb M-MMn steel. In this case, subsequent partitioning or tempering at 250 to 350 °C slightly increases the carbide size without major changes in morphology. Resultantly, the partitioning increases the yield stress and slightly decreases the tensile strength with no change in elongation.

Hojo et al. [82] show that warm-forging at 650 °C and subsequent slow cooling (1 °C/s) increases the volume fraction and carbon concentration of the retained austenite and decreases the volume fractions of MA phase and carbide in 0.23%C–1.5%Si–1.5%Mn–1.0%Cr–0.2%Mo–0.1%Ti–0.0019%B TM steel (Figure 18). The product of shear stress and shear elongation of the TM steel after warm-forging, which is measured by small punching tests at room temperature, is slightly increased.

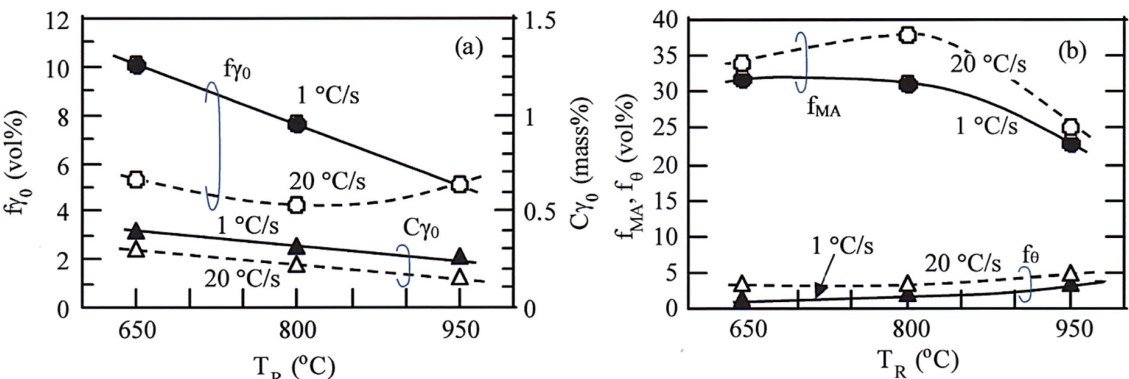

**Figure 18.** Variations in (**a**) initial volume fraction ($f\gamma_0$) and initial carbon concentration ($C\gamma_0$) and (**b**) volume fractions of MA phase ($f_{MA}$) and carbide ($f_\theta$) of retained austenite as a function of warm working temperature ($T_R$) in 0.23%C–1.5%Si–1.5%Mn–1.0%Cr–0.2%Mo–0.1%Ti–0.0019%B TM steel [82]. Cooling rate: 1 °C/s (solid marks) or 20 °C/s (open marks). Reduction strain: 40%. Reprinted with permission from AIST: Iron Steel Technol., Copyright 2021.

## 7. Case Hardening of AMSs

Fine particle peening (FPP) treatment after the heat-treatment (the DQ-P process, $T_p = 180$ °C) increases the rotating bending fatigue limits of smooth and notched specimens in 0.20%C–1.50%Si–1.51%Mn–1.0%Cr–0.05%Nb TM steel (Figure 19a) [147]. Vacuum carburization (VC) treatment followed by the DQ-P process and subsequent FPP treatment (VC+FPP, $T_p = 180$ °C) further increases the fatigue limits in 0.2%C–1.5%Si–1.5%Mn–1.0%Cr–0.05%Nb TM steel [148]. According to Sugimoto et al., the high fatigue limits are obtained under the conditions of carbon potential of 0.8 mass% (Figure 19b) [148] and arc-height of 0.21 mm with a gage N (Figure 19a) [147].

The fatigue limits of case-hardened TM steel are mainly controlled by the Vickers hardness (or yield stress) and compressive residual stress in the surface hardened layer, which are developed from the severe strain-hardening and strain-induced transformation of retained austenite [147,148]. As shown by line (1) in Figure 20, the smooth fatigue limit ($\sigma_w$) of 0.2%C–1.5%Si–1.5%Mn–1.0%Cr–0.05%Nb TM steel subjected to only the FPP treatment shows a linear relationship with the sum of the estimated maximum yield stress

and the absolute value of maximum compressive X-ray residual stress ($\sigma_{Y,est} + |\sigma_{X\alpha,max}|$) as the following equation proposed by Matsui and Koshimune et al. [149,150],

$$\sigma_w = 0.3891 \times (\sigma_{Y,est} + |\sigma_{X\alpha,max}|), \quad (2363\ \text{MPa} < \sigma_{Y,est} + |\sigma_{X\alpha,max}| < 4505\ \text{MPa}) \quad (10)$$

where $\sigma_{Y,est}$ is calculated by

$$\sigma_{Y,est} = (HV_{max}/3) \times 9.80665 \times (YS/TS) \quad (11)$$

where $HV_{max}$ is the maximum Vickers hardness and YS/TS is the yield ratio (assumed by Matsui et al. to be 0.95 in 0.22%C–0.27%Si–0.74%Mn–1.1%Cr–0.36%Mo Q&T steel gas-carburized and then shot-peened [150]).

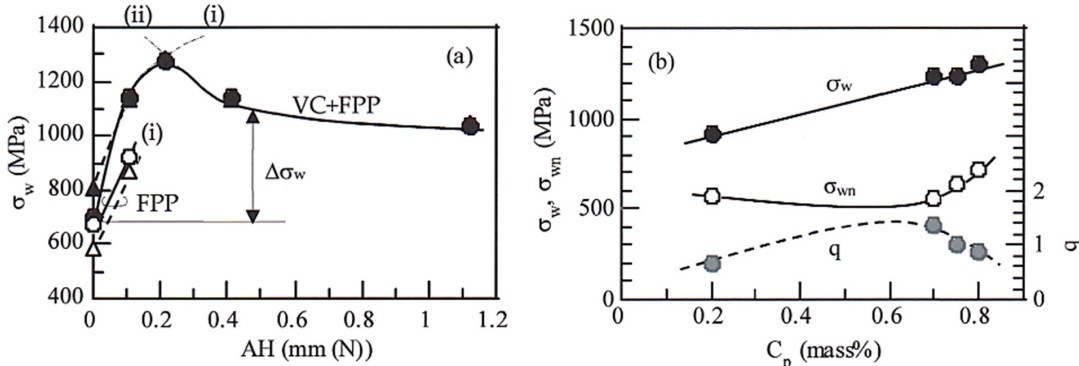

**Figure 19.** (**a**) Arc-height (AH) dependence of smooth fatigue limit ($\sigma_w$) in 0.2%C–1.5%Si–1.5%Mn–1.0%Cr–0.05%Nb TM steel (●○) and JIS-SNCM420 Q&T steel (△▲) subjected to fine particle peening (FPP) treatment after heat-treatment (the DQ-P process, $T_p$ = 180 °C) and vacuum carburization (VC) followed by the DQ-P process ($T_p$ = 180 °C) and then fine particle peening (FPP) treatment (VC+FPP) [147]. (**b**) Carbon potential ($C_p$) dependence of $\sigma_w$, notch-fatigue limit ($\sigma_{wn}$) and notch sensitivity ($q$) in the TM steel subjected to VC+FPP treatment [148]. (i) Increases in hardness and compressive residual stress, (ii) fish-eye crack fracture at a high depth. Refs. [147,148] are reprinted with permissions from Springer Nature: Metall. Mater. Trans. A, Copyright 2021 and from Taylor & Francis: Mater. Sci. Technol., Copyright 2021, respectively.

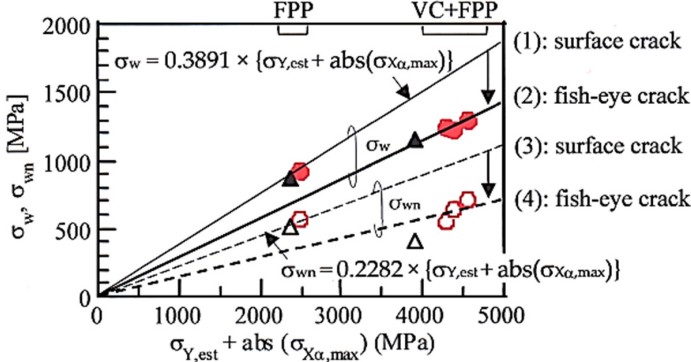

**Figure 20.** Relationships between smooth and fatigue limits ($\sigma_w$, $\sigma_{wn}$) and the sum of estimated yield stress and the absolute value of maximum compressive residual stress {$\sigma_{Y,est} + abs(\sigma_{X\alpha,max})$} of smooth (solid marks) and notched (open marks) specimens in 0.2%C–1.5%Si–1.5%Mn–1.0%Cr–0.05%Nb TM (●○) and JIS-SNCM420 (▲△) steels subjected to only fine particle peening (FPP) with an arc height of 0.21 mm(N) after the heat-treatment (DQ-P process, $T_p$ = 180 °C)) and vacuum carburization under various carbon potentials ($C_P$ = 0.2 to 0.8 mass%) and subsequent FPP (VC+FPP). Lines (1) and (3) denote the fatigue limits of the smooth and notched specimens subjected to FPP treatment, respectively. Lines (2) and (4) refer to the fatigue limits of the smooth and notched specimens subjected to VC+FPP treatment [148]. Reprinted with permission from Taylor & Francis: Mater. Sci. Technol., Copyright 2021.

As shown in Figure 20, the slope in Equation (10) of notch-fatigue limit ($\sigma_{wn}$) lowers to 0.2282 in TM steel subjected to the FPP treatment (see Line (3)) [147]. On the other hand, the slops corresponding to the smooth and notch-fatigue limits of TM steel subjected to the VC+FPP treatment further decrease such as Lines (2) and (4), respectively [147,148]. The decreased slopes of Lines (2) and (4) are caused by the fish-eye crack fracture. It is noteworthy that the smooth and notch-fatigue limits of TM steel are higher than those of JIS-SNCM420 Q&T steel subjected to the same process.

Skołek et al. [151] report that DIN-35CrSiMn5-5-4 steel subjected to the DQ-P (or Q&T) process after the VC treatment (without FPP or shot-peening) possesses low surface wear resistance compared to the steel with nano-bainitic structure subjected to the IT process above $M_s$. They say that the low wear resistance of DQ-P process steel is caused by low retained austenite fraction. Kanetani et al. [152] investigate the morphology of the deformation-induced martensite formed on rolling contact fatigue in SAE4320 steel subjected to carburization and then Q&T process and show an interesting result that the deformation-induced martensite is formed preferentially within the retained austenite grains, not from the interface between tempered martensite and retained austenite.

## 8. Summary

The AMS sheets can be produced by lower heat-treatment temperature than the third-generation AHSSs (Type A) with BF/M structure matrix. As they possess the same excellent cold formability as the third-generation AHSSs (Type A), some applications to the automotive frame members can be expected in the future [153–155]. In addition, the AMSs can be applied to the automotive wire rod and bar parts such as engine, powertrain and chassis parts. Thus, the wear property, weldability, and machinability of the hot-forged AMSs also must be systematically investigated in the future, apart from toughness, fatigue strength, and delayed fracture strength described in this review. In addition, the effects of case-hardening on the mechanical properties are also required, because the engineers and scientists want to know them.

Finally, it is emphasized that the AMSs may be applied to not only automotive parts but also the forging parts of other engineering structures such as construction machinery, airplane, marine machinery, etc. Such applications will increase the fracture strength of many products and resultantly bring about a great increase in reliability. The microstructure and mechanical properties of martensitic stainless steel containing metastable retained austenite subjected to the DQ and DQ-P process should be also discussed as one kind of AMSs in the future.

**Funding:** This research received no external funding.

**Institutional Review Board Statement:** Not applicable.

**Informed Consent Statement:** Not applicable.

**Data Availability Statement:** Not applicable.

**Acknowledgments:** I thank Tomohiko Hojo from Tohoku University and Junya Kobayashi from Ibaraki University for their kind discussion.

**Conflicts of Interest:** The author declares no conflict of interest.

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
