# Peer review of "Recent Progress of Low and Medium-Carbon Advanced Martensitic Steels"

_metals, doi:10.3390/met11040652_

Round 1

Reviewer 1 Report

Review for metals- 1183125

Recent Progress of Low and Medium Carbon Advanced Martensitic Steels

The authors address an interesting research topic for the journal Metals. In my opinion, this review is an instructive, very didactic, and useful study. In addition, it is rigorous and well-organized work. However, for the sake of clarity, I would recommend the authors to improve the explanation of Figures 4, 9, and 13b.

Author Response

The authors address an interesting research topic for the journal Metals. In my opinion, this review is an instructive, very didactic, and useful study. In addition, it is rigorous and well-organized work. However, for the sake of clarity, I would recommend the authors to improve the explanation of Figures 4, 9, and 13b.

[Reply]

Thanks so much for your kind comment. The author revised as follows according to the reviewer’s comment.

It is difficult to improve this text. The author added the “blocky γR” and “filmy γR” in Figs. 4, 9, and 13b.

Reviewer 2 Report

Dear Authors,

I have read your paper titled "Recent Progress of Low and Medium Carbon Advanced Martensitic Steels".

This manuscript fulfills the aims and scope of the Metals. I have some comments and suggestions, which are listed below.

My overal merit about your work is high. You have presented relevant investigations to write Review type of the paper. Congratulations. I propose minor revision.

In my opinion you should describe gaps in knowledge more deeply.

Introduction:

  • Fig. 1 - I propose to show this figure bigger.
  • Please describe wider the usage of investigated materials.

Two kinds of heat-treatment process forAï¼­Ss:

  • Fig. 2 - the unit of scale bars is missing. Also, the some values should be presented. Potential readers should know in which range of values we are focused.

Microstructure and retained austenite characteristics of AMSs:

  • Ref (1) needs reference.

Tensile propertiesand cold formability of AMS sheets:

- This section is clear.

Mechanical Properties of heat-treated AMS plates andbars:

- Fig. 8, Fig. 10 - the resolution should be improved.

Hot-and warm-workingsto produceautomotiveAMS parts:

  • Fig. 17 - Please check the area of "bainite"- the white pixels could be observed. Please improve this issue.

Case hardening of AMS:

- This section is clear.

Summay:

In my opinion, you should describe the gaps of knowldge here. AAlso you should mark the ways of future investigations.

Author Response

I have read your paper titled "Recent Progress of Low and Medium Carbon Advanced Martensitic Steels".

This manuscript fulfills the aims and scope of the Metals. I have some comments and suggestions, which are listed below.

My overall merit of your work is high. You have presented relevant investigations to write the Review type of the paper. Congratulations. I propose minor revision.

In my opinion you should describe gaps in knowledge more deeply.

[Reply]

Thanks so much for your kind review. The author revised the manuscript according to the reviewer’s comments, as follows.

  1. Introduction:
  • 1 - I propose to show this figure bigger.
  • Please describe wider the usage of investigated materials.

[Reply]

  • 1 was made bigger.
  • The following usage was added to line 53.

For the third-generation AHSSs (Type A), a TS×TEl higher than 30 GPa% is required to apply to the automotive body frame members, the seat frame members, the concrete mixer truck cylinders, etc.

  1. Two kinds of heat-treatment process for Aï¼­Ss:

Fig. 2 - the unit of scale bars is missing. Also, the some values should be presented. Potential readers should know in which range of values we are focused.

[Reply]

To understand the heat-treatment process in this figure, transformation temperatures such as Ac3, Ms, and Mf are important because they are varied by the chemical composition of the steel, as well as room temperature. Therefore, the author omits the unit of scale bars.

  1. Microstructure and retained austenite characteristics of AMSs:

Ref (1) needs reference.

[Reply]

Unfortunately, I do not understand the above comment.

  1. Tensile properties and cold formability of AMS sheets:

- This section is clear.

  1. Mechanical Properties of heat-treated AMS plates and bars:

- Fig. 8, Fig. 10 - the resolution should be improved.

[Reply]

The sharpness and contrast of Figs. 8 and 10 were modified.

  1. Hot-and warm-workings to produce automotive AMS parts:

Fig. 17 - Please check the area of "bainite"- the white pixels could be observed. Please improve this issue.

[Reply]

Thanks. The author revised slightly the area of “bainite”. Also, I moved the white pixels.

  1. Case hardening of AMS:

- This section is clear.

8. Summay:

In my opinion, you should describe the gaps of knowldge here. Also you should mark the ways of future investigations.

[Reply]

Author modified the summary according reviewer’s comment as follows,

The AMS sheets can be produced by lower heat-treatment temperature than the third-generation AHSSs (Type A) with BF/M structure matrix. As they possess the same excellent cold formability as the third-generation AHSSs (Type A), some applications to the automotive sheet parts frame members can be expected in the future [153−155]. Also, the AMSs can be applied to the automotive wire rod and bar parts. Thus, the wear property, weldability and machinability of the hot-forged AMSs also must be systematically investigated in the future, as well as apart from toughness, fatigue strength and delayed fracture strength described in this review. Besides, the effects of case-hardening on the mechanical properties are also required because the engineers and scientists want to know them.

Reviewer 3 Report

The topic of modern martensitic steels is topical. The manuscript submitted for review is well organized and carefully divided into sub-sections. The author cites current works. Most of the works cited are younger than 2010, which makes this review article current. The author does not describe the processes in great detail, e.g. hot stamping, but gives the reader a signal so that the reader can find more information. The theme is well chosen. The author has extensive experience in this topic. Perhaps thanks to this, the work is well prepared. 

Author Response

The topic of modern martensitic steels is topical. The manuscript submitted for review is well organized and carefully divided into sub-sections. The author cites current works. Most of the works cited are younger than 2010, which makes this review article current. The author does not describe the processes in great detail, e.g. hot stamping, but gives the reader a signal so that the reader can find more information. The theme is well chosen. The author has extensive experience in this topic. Perhaps thanks to this, the work is well prepared.

[Reply]

Thanks so much for your kind review.

Reviewer 4 Report

This manuscript meets the publication requirements